# The Surface Behavior of ZnO Films Prepared at Room Temperature

Da-Hua Wei [1],*, Sheng-Kai Tong [2], Sheng-Chiang Chen [1] and Rong-Tan Huang [3],*

[1] Institute of Manufacturing Technology, Department of Mechanical Engineering, National Taipei University of Technology (TAIPEI TECH), Taipei 10608, Taiwan

[2] Research and Development Department, CB-CERATIZIT Group, New Taipei City 24250, Taiwan

[3] Department of Optoelectronics and Materials Technology, National Taiwan Ocean University, Keelung 20224, Taiwan

* Correspondence: dhwei@ntut.edu.tw (D.-H.W.); rthuang@mail.ntou.edu.tw (R.-T.H.)

**Abstract:** The surface behavior of ZnO-based films can be modulated via the postannealing and ultraviolet (UV) illumination of different strengths and durations, respectively. The present results could provide the basis for modulating their microstructures with respect to the grain-size distribution and photocatalytic behavior, and act as a potential guide in the field of wide-bandgap semiconducting oxides. ZnO films were prepared at room temperature onto Corning-1737 glass substrates by applying radio-frequency magnetron sputtering without supplying an oxygen source. With the purpose of obtaining modulational grain microstructures, the as-prepared ZnO films (Z0) were treated via a postannealing modification in a vacuum furnace at 300 °C for 30 min after deposition (Z300), accompanied by adjustable internal stress. The contact angle (CA) value of the ZnO films was reduced from 95° to 68°, owing to the different grain microstructure accompanied by a change in the size variation. In addition, UV light with different illumination strengths could be used to improve the hydrophilicity, which varied from a hydrophobic status to a superhydrophilic status due to the desirable surface characteristics of its photocatalytic action. In addition, the photocatalytic activity of the ZnO films exhibited an effectual photodegradation of methylene blue (MB) under UV illumination, with a chemical reaction constant of $2.93 \times 10^{-3}$ min$^{-1}$. In this present work, we demonstrated that the CA value of the ZnO films not only caused a change from a hydrophobic to hydrophilic status, accompanied by a change in grain size combined with internal stress, but also, induced by the UV light illumination, was combined with photocatalytic activity simultaneously. On the other hand, an enhanced surface plasmonic resonance was observed, which was due to couple oscillations between the electrons and photons and was generated from the interface by using a flat, continuous Pt capping nanolayer. This designed structure may also be considered as a Pt electrode pattern onto ZnO (metal Pt/ceramic ZnO) for multifunctional, heterostructured sensors and devices in the near future.

**Keywords:** surface behavior; internal stress; ultraviolet light strength; photocatalytic; heterostructure

## 1. Introduction

The surface behavior of oxide materials has been considered for various powerful applications, including sensors, detectors, the cleanup of the environment, energy conversion, and related memory nanodevices [1–12]. Various sensors and detectors have been focused on and developed as one-dimensional (1-D) nanostructures, presenting a difficult task for their super large-scale manufacture with the required uniformity. Functional coatings can be deposited onto metal, ceramics, and semiconductor bottom-based materials, and their bottom-up nanostructures are an easy and useful way of significantly enhancing the surface plasmonic performance in electrical and optical applications [13–17]. Thus, strong plasmon–exciton coupling in a metallic capping shell combined with semiconductor-based materials

has potentially provided a method for enhancing the plasmon–exciton interaction and its corresponding specificity in many application areas [18–20]. Hydrophobic/hydrophilic functional coatings are usually fabricated by modulating the surface status with a low/high surface energy for generating nano/microstructures. It is easier to form a texture with zinc oxide (ZnO), with its lowest surface energy at the (002) plane; also, ZnO is the most commonly used material for photocatalytic reactions among the metal oxides. ZnO compounds display a good photoreactivity under ultraviolet (UV) light illumination, owing to their wide bandgap of 3.37 eV [21], and their surface wetting state can be modulated significantly by the illumination strength of light energy from UV [22–26]. Therefore, a semiconducting ZnO compound is a suitable selection for a multifunctional surface coating. A reversible UV light-driven hydrophobic/hydrophilic surface characteristic for ZnO-based nanostructures and related nanowires/rods decorated with metallic nanoparticles has been widely developed and demonstrated [27–30].

Semiconducting ZnO compounds typically have exciton binding energy, a direct wide bandgap, and are highly transparent in the visible region, with a transmittance of over 80%. Hence they are an excellent selection for future potential applications and are used in UV detectors, light emitters, transparent thin film transistors, solar cells, piezoelectric transducers, chemical gas sensors, biosensors, and resistive random-access memory (RRAM) devices [31–36]. Owing to its many possible applications, ZnO has a modulational effect in its surface characteristics, which can provide outstanding surface behavior, including photocatalytic activity and reversible hydrophobic/hydrophilic wetting transition via UV light illumination, originating from its intrinsic semiconducting characteristics. The reversible wettability and modulation of ZnO compounds are due to the competing surface chemical characteristics of desorption and adsorption and the resulting rearrangement of organic chains and hydroxyl groups on their surface status [37,38]. Photoreactive materials such as ZnO have been highly applied for their surface characteristics and used for their display of modulated wetting behavior, combined with their quick hydrophobic/hydrophilic transition with modulational contact angles. Various methods have claimed to be successful in controlling surface microstructures, which have been developed to modulate the wetting and photocatalytic behavior of functional coatings. On the other hand, different skills for the growth of ZnO hetroepitaxy films have been reported [39–47]; the sputtering coating skill is a suitable one for manufacturing large-area uniform ZnO films targeted for use in the future for cleaning up the environment, related optoelectronic nanodevices, and biosensors, because the process has received important attention due to its rapid growth of high-quality coatings with stable deposition rates at a low temperature. It can be understood that the deposition uniformity of nanostructures, thin, and thick films with super-large areas is controlled by many sputtering conditions, including the sputtering power, the sputtering pressure, the inset gas supply, the substrate temperature, and the distance between the target and the substrate [48,49].

In this present work, the surface behavior of highly textured ZnO films, prepared by applying the RF sputtering system without supplying an oxygen source at room temperature, was investigated. The wetting behavior, grain size, and morphological features on the surfaces of the textured ZnO films were modulated via a direct postannealing modification. The transmittance for all the textured ZnO films was also examined and demonstrated. The surface wetting behavior of the textured ZnO films was checked using water contact angle measurement equipment. The modulational wetting behavior was obtained by adjusting the strength of the UV light illumination and storage in a dark room for 12 h in order to reach the initial water contact angle, which was contributed from the photocatalytic reaction. Our claimed work also offers a simple skill for modulating the wetting behavior of ZnO films by modulating the UV power density. The coupling of surface plasmons and excitons using a continuous Pt capping nanolayer onto a ZnO film was selected to study the coupling effect for the metal/semiconducting oxide heterostructures and could be used for the fabrication of nanodevices in future work. Our proposed work presents a convenient and original method for modulating surface behavior and enhancing surface plasmons



for wide-bandgap semiconducting ZnO, just by directly capping a metallic nanolayer not alike to decorate with metallic nanoparticles. This not only enlarges the future demands of ZnO compounds, but also power assists a direct way with a wide-bandgap semiconducting material sketch and nanodevice process on flexible and transparent substrates for future wearable technologies.

## 2. Experiments and Film Structures

All the textured ZnO films were prepared at room temperature onto Corning-1737 glass substrates by applying radio-frequency (RF) magnetron sputtering without supplying an oxygen source. The glass substrates were placed in a chamber parallel to a commercial ZnO binary target with a 99.99% purity, whose thickness and diameter were 3 mm and 51 mm, respectively. The glass substrates were washed in acetone and ethanol to remove organic contaminants at first, then immersed in deionized water ultrasonically, and finally the glass substrates were set into the working chamber after being dried in hot air. The high-vacuum working chamber was pumped down to a base pressure of $4.5 \times 10^{-7}$ torr. Argon gas was then used in the working chamber with a working pressure of $1 \times 10^{-2}$ torr, without supplying oxygen gas. The ZnO films were prepared using 75 W RF power, and all the film thicknesses were fixed at about 250 nm. Furthermore, with the purpose of obtaining modulational grain microstructures, the as-prepared ZnO samples (Z0) were treated with a postannealing modification in furnace at 300 °C for 30 min after deposition (Z300), respectively.

The crystallinity and related crystalline preferred orientation of the ZnO films was performed using an X-ray diffraction analysis (XRD, PANalytical, Almelo, The Netherlands) with Cu Kα radiation (λ = 1.54 Å), via the 2θ ranges of 20–90°. The morphological features of the surfaces and related elemental information of the ZnO films were performed using a scanning electron microscope (SEM, Phenom XL G2, Thermo Scientific, Waltham, MA, USA) and field emission scanning electron microscopy (FE-SEM, Dresden, Germany), respectively. The chemical stoichiometry of the ZnO compounds was identified to be $Zn_{40}O_{60}$ using a field emission electron probe X-ray microanalysis (FE-EPMA, JEOL, Tokyo, Japan). The contact angle measurement and calculation for the relating surface free energy of the ZnO films were obtained from the water contact angle (CA, Rame-Hart 100 goniometer, Capovani Brothers Inc., Scotia, NY, USA) measurement by using a small drop of liquid water onto the surface of an individual ZnO sample. The degree of accuracy for the water contact angle measurement was induced by the image quality, using a charge-coupled device (CCD) to capture the images of the drop of liquid water (~5 μL). Then, the processing of the fitting curve function was achieved immediately by using CA software, and the value was evaluated to be about ±1 degree. In addition, the related measurement of the surface free energy (SFE) for an individual sample was also evaluated at the same time. The power energy of the ultraviolet (UV) light illuminated with the ZnO films was between 330 and 3330 mW/cm$^2$, induced by using a mercury arc lamp (HAMAMATSU-Deuterium L2D2) with a 365 nm wavelength. In this work, the as-prepared ZnO films without any postannealing modification and with a 300 °C postannealing modification will be denoted below as Z0 and Z300, respectively.

## 3. Results and Discussion

The X-ray diffraction patterns for the Z0 and Z300 samples are shown in Figure 1a. The XRD spectra confirm that both samples demonstrated a tough peak positioned at approximately 2θ = 34°, which was related to the high grade of the *c*-axis orientation with a wurtzite crystal structure along the ZnO (002) plane (JCPDS Card: 36-1451). The great intensity of the diffraction spectra generated from the ZnO (002) plane was owed to the puniest surface energy in the ZnO (002) basal plane in the wurtzite crystal structure, promoting a crystallographic orientation together with the [001] preferred crystalline direction. The peak intensity of the ZnO (002) diffraction angle was enhanced considerably by increasing the temperature and time of the postannealing modification, which was due

to much more energy improving the crystallinity via the postannealing process. In addition, in the XRD spectrum of sample Z300, there is still one slight peak located at around 76.6°, indexing as ZnO (004). It can be known that, without any peaks except that at (00n), diffraction angles were obtained over a large angular range (θ–2θ scan), indicating the ZnO film did form onto the glass substrates epitaxially. On the other hand, it was demonstrated that the ZnO (002) diffraction peak slightly changed from a low-angle region (2θ = 34.36°) toward a high-angle region (2θ = 34.39°), as shown in Figure 1b, indicating that the internal stress of the ZnO films was modulated via a direct postannealing modification.

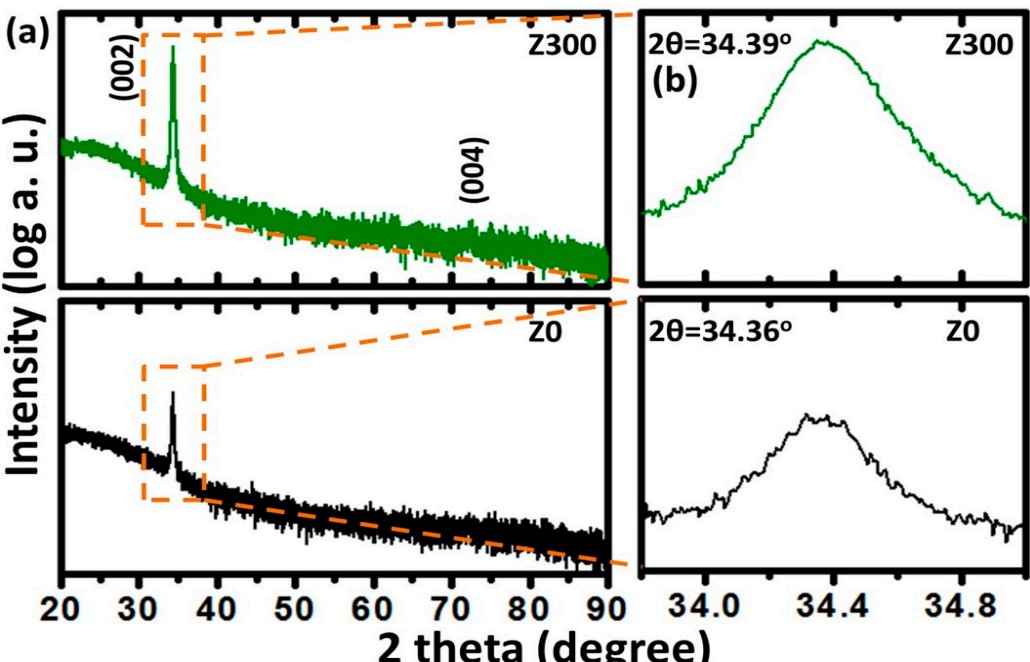

**Figure 1.** (**a**) X-ray diffraction pattern for the as-prepared ZnO films (Z0) that were then modulated via postannealing in vacuum furnace at 300 °C for 30 min after deposition (Z300), respectively. (**b**) The relating slow-scan spectra for the ZnO (002) diffraction peak in the θ–2θ region of (**a**).

With the purpose of confirming the internal stress between these two samples, the internal stress σ of the hexagonal wurtzite crystal structure is illustrated as [50]:

$$\sigma = 450(C - C_0)/C_0 \text{ GPa} \tag{1}$$

where C is the lattice parameter/constant for the *c*-axis of the hexagonal crystal structure that can be obtained from the tough ZnO (002) peak originated from the XRD patterns, and $C_0$ is 5.206 Å, the corresponding lattice constant value for the bulk ZnO. As shown in Equation (1), the lattice constant $C_0$ is used to calculate the internal stress of the textured ZnO compound. Furthermore, for the purpose of estimating the lattice constant, Bragg's equation can be used for determining the lattice spacing of the crystals.

$$n\lambda = 2d \sin\theta \tag{2}$$

In this case, λ is the shortest wavelength of an incident X-ray (λ = 1.54 Å) and $d_{(002)}$ is for the interplanar spacing of the hexagonal wurtzite structure of the ZnO (002) facet. As shown in Figure 1, the ZnO (002) diffraction peaks of both samples are located at 2θ = 34.36° (Z0) and 34.39° (Z300), respectively. After substituting θ into Equation (2), the $d_{(002)}$ of each sample are 2.607 Å (Z0) and 2.605 Å (Z300), respectively. Therefore, based on the above typical definitions, the *c*-axis lattice constant of ZnO can be calculated from the above obtained results. Therefore, the values of the lattice constant for each sample are 5.214 Å (Z0) and 5.21 Å (Z300), respectively. At last, by replacing each lattice constant in

Equation (1), the internal stress for the textured ZnO films could be calculated and obtained. As shown in Figure 2, the internal stresses of samples Z0 and Z300 were calculated using the obtained information from the X-ray diffraction spectra, and the calculated values of the internal stresses for samples Z0 and Z300 were −0.76 GPa and −0.46 GPa, respectively. Apparently, the values of the internal stresses for Z0 and Z300 were indeed modulated by a simple postannealing modification. According to the obtained results above, the internal stress of the textured ZnO films was modulated via a suitable postannealing temperature and time; hence, this kind of transition was attributed to the much higher heat energy occurring during the postannealing modification. Thus, the energetic particles bombardment and design of the heterostructured films could be applied to the laxation of the internal stress and accompanied by the transition of the grain-size morphology [51,52].

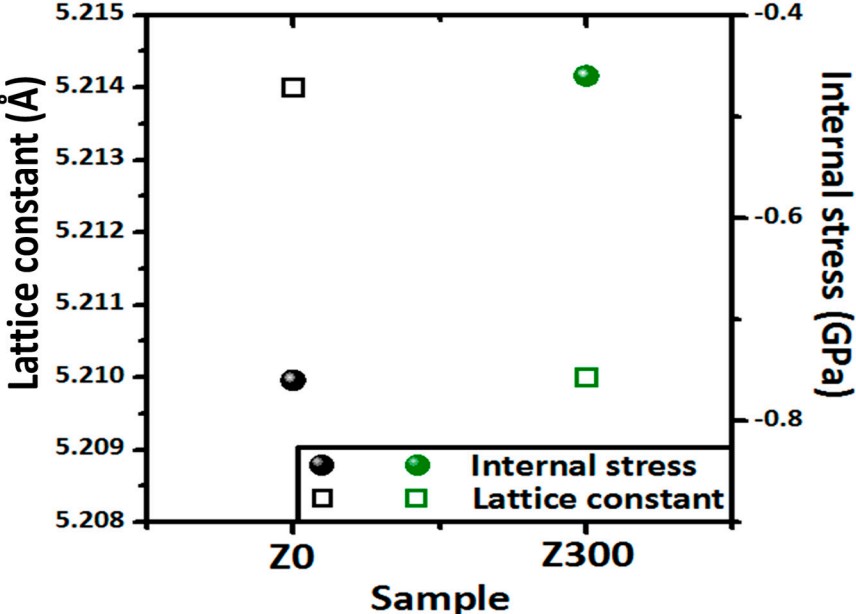

**Figure 2.** The relationship between lattice parameter/constant and internal stress of samples Z0 and Z300, respectively.

Meanwhile, the different internal stresses would result in different microstructures, and the plan-view SEM images for the as-prepared ZnO films and those with a postannealing modification at 300 °C are shown in Figure 3a,b. Figure 3a,b display the modulational grain microstructures accompanied by a change in the grain size corresponding to samples Z0 and Z300, respectively. This kind of grain size transition was induced by a postannealing modification that could be easily manipulated. The as-prepared ZnO film's (Z0) average grain size was 29 ± 2 nm, as displayed in the inset of Figure 3a. In addition, the average grain size of the ZnO film with a postannealing modification at 300 °C (Z300) was 53 ± 2 nm, as shown in the inset of Figure 3b. The grain size histograms for the fractions and distribution analysis using a Gaussian mode fitting curve (red line) are shown in the inset of Figure 3a,b. Such a kind of improvement could be attributed to obtaining a much higher heat energy during the postannealing modification that directly led to an enhancement in the grain growth kinetics, and the microstructure of the ZnO films could be modulated via the varied temperature and time of the postannealing modification—this could be known via the postannealing modification. The nanograins with complex domains amalgamated and combined with each other to develop grain coalescence, as displayed in Figure 3b. This could also have been due to the presence of apertures or pores located between the grains that could absorb oxygen inside and exist on the surface of film. The corresponding images of the water droplets on the ZnO surfaces are displayed in Figure 3c,d. The surface water contact angles (CAs) were 95° and 68° for the Z0 and Z300 samples, respectively. The as-prepared ZnO film showed a hydrophobic wetting status, owing to

the smaller nanograin size, and much more apertures supplied the trapping of oxygen that decreased the contact area between the surface area of the ZnO and water droplets, as displayed in Figure 3c. With a postannealing modification at 300 °C, the microstructure surface showed a hydrophilic status, owing to the giant grain size and smooth surface, so it would obtain smaller apertures that could not supply the trapping of oxygen between the interface of the ZnO compound and the water droplet, as shown in Figure 3d. This transformation caused the CA to decrease from a hydrophobic (CA = 95°) to hydrophilic (CA = 68°) status, and it was induced by the fewer apertures formed with a postannealing modification at 300 °C. The schematic diagram illustrates the different modes, displaying the status of the water droplets on the different surface structures of the ZnO compounds, as shown in Figure 3e. The microstructure containing smaller grains had much more apertures and exhibited a hydrophobic wetting status without any modification. In addition, the smooth surface with a grain growth phenomenon led to far less oxygen trapping between the interface of the ZnO compound and the water droplet, and then induced the wetting behavior to change from a hydrophobic to hydrophilic status, which served as a surface indication of the grain coalescence to form a giant grain size. In short, the surface hydrophilicity was promoted when there was less oxygen absorbed between the solid and liquid surface, demonstrating a decrease in CA via a simple postannealing modification, as schematically displayed in Figure 3e.

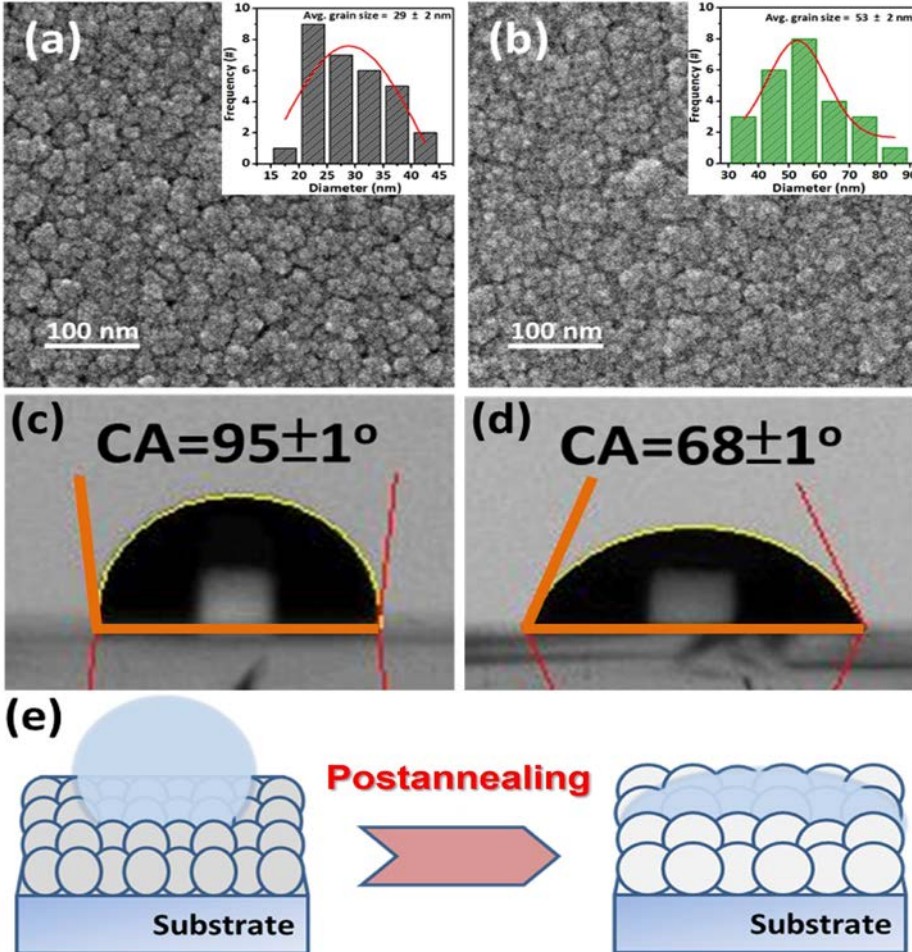

**Figure 3.** Plan-view FE-SEM images and the insets showed calculated value of the average grain size with Gaussian mode fitting curve (red line) and its distribution are for the different samples, (**a**) Z0 and (**b**) Z300, respectively. (**c**,**d**) Relating contact angle images for samples Z0 and Z300, respectively. (**e**) Schematic diagram is a transition of surface wetting behavior for ZnO samples without and with a postannealing modification.

Maintaining a high level of visible transmittance of a device acts a key role in particular and industrial applications, such as action wearable devices and the panels of solar cells. Figure 4 shows the optical transparency for the as-prepared and postannealed, modified ZnO compounds, respectively. From the spectra of the optical transparency, the Z0 and Z300 samples exhibited a high transmission in the visible region, with average transmittance values of 83.6% and 81.7%, respectively. The average transmittance of sample Z300 was slightly less than sample Z0, and this was attributed to the much more flawed formation, including mainly oxygen vacancies, which were formed while the oxygen gas was lost under the vacuum atmosphere during the postannealing modification. From the above-mentioned results, all the highly textured ZnO films had an excellent optical transparency and exhibited a visible light transmittance beyond 80%. It is clearly displayed that both hydrophobic and hydrophilic surface wetting statuses of ZnO compounds have the potential to be applied to smart windows combined with the related optoelectronic, semiconducting, and energy nanodevices.

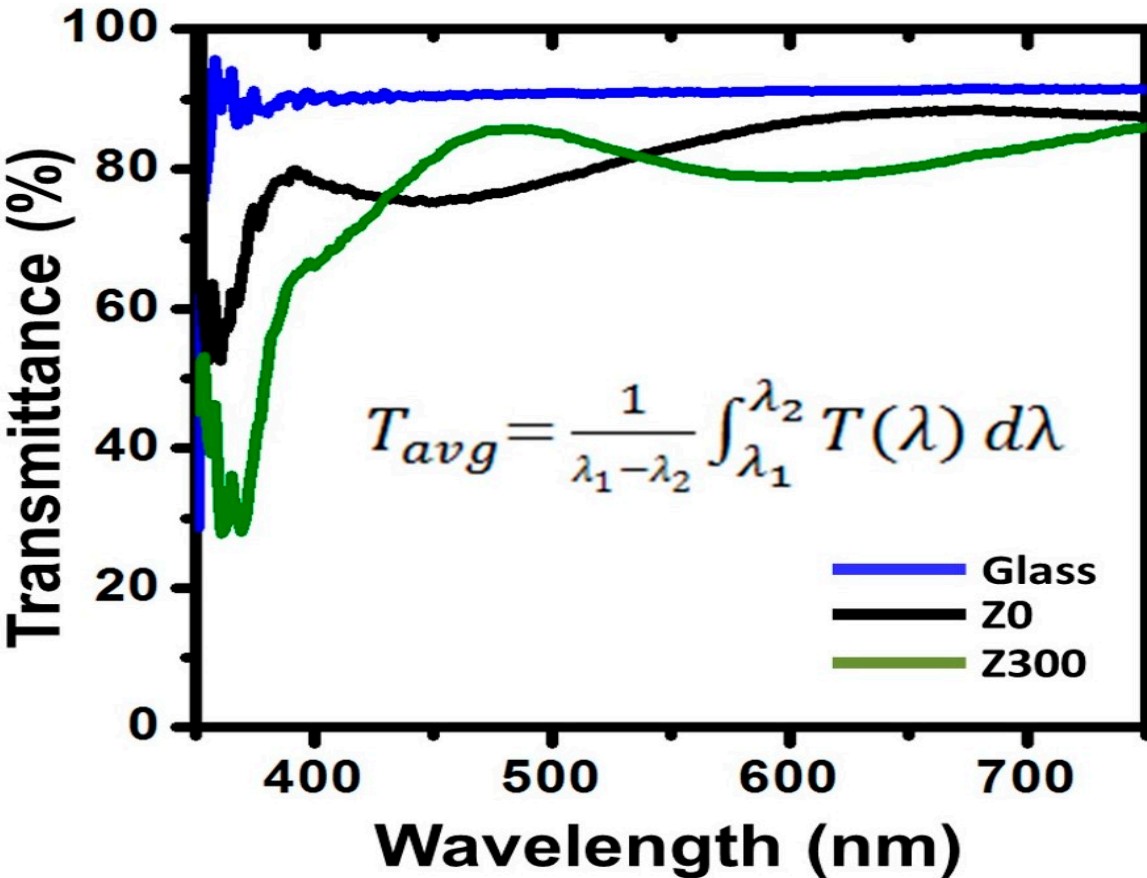

$$T_{avg} = \frac{1}{\lambda_1 - \lambda_2} \int_{\lambda_1}^{\lambda_2} T(\lambda)\, d\lambda$$

**Figure 4.** Transmittance spectra of the prepared samples Z0 and Z300 compared with pure glass substrate, respectively.

As a photonic material, it is important to verify the optical performance of ZnO compounds, and photoluminescence (PL) spectra are considered as a regular method. Generally, it is known that the PL emission spectrum of ZnO compounds can be separated with two main sections. The first section belongs to the near-band-edge (NBE) emission spectrum, which is originated from the free-exciton recombination generation in the UV zone. Another one is the deep-level (DL) emission spectrum, which is originated from the possible defects and related impurities, mainly including oxygen vacancies and zinc interstitials that are caused by the suppression of the DL emission spectrum in the visible zone. Figure 5 displays the PL spectra for both samples Z0 and Z300, respectively. Both samples demonstrate a tough NBE emission spectrum, which is positioned at approximately

374 nm and acts as the dominant recombination mechanism of photogenerated charge carriers (excitonic photoemission). In contrast, another negligible defect band, which is positioned at approximately 546 nm (DL emission), induced by the intrinsic defects, mainly including oxygen vacancies and zinc interstitials, possibly existed in the ZnO compound, demonstrating that the ZnO compounds were nearly defect-free. It is noteworthy that the tough peak of the NBE emission spectrum for sample Z300 was slightly changed toward a high-wavelength band compared to the sample Z0. Based on previous published works, the peak shifting of the NBE emission for the PL spectrum of the ZnO in our present work can be induced by the quantum-size confinement effect [53–55]. On the other hand, the PL peak intensity in the UV band of sample Z300 was slightly larger and narrower than the as-prepared sample Z0, which was related to the degree of crystallinity in the ZnO compound. In other words, it indicates that the PL spectra display an identical trend compared to the analysis results of the XRD, as both the crystallinity and optical characteristics were simultaneously enhanced. Furthermore, the PL intensity of the peak in the DL band with sample Z300 was slightly greater than that of sample Z0, owing to the increasing oxygen vacancy while losing the oxygen atoms from the ZnO compound during the postannealing modification under a vacuum atmosphere.

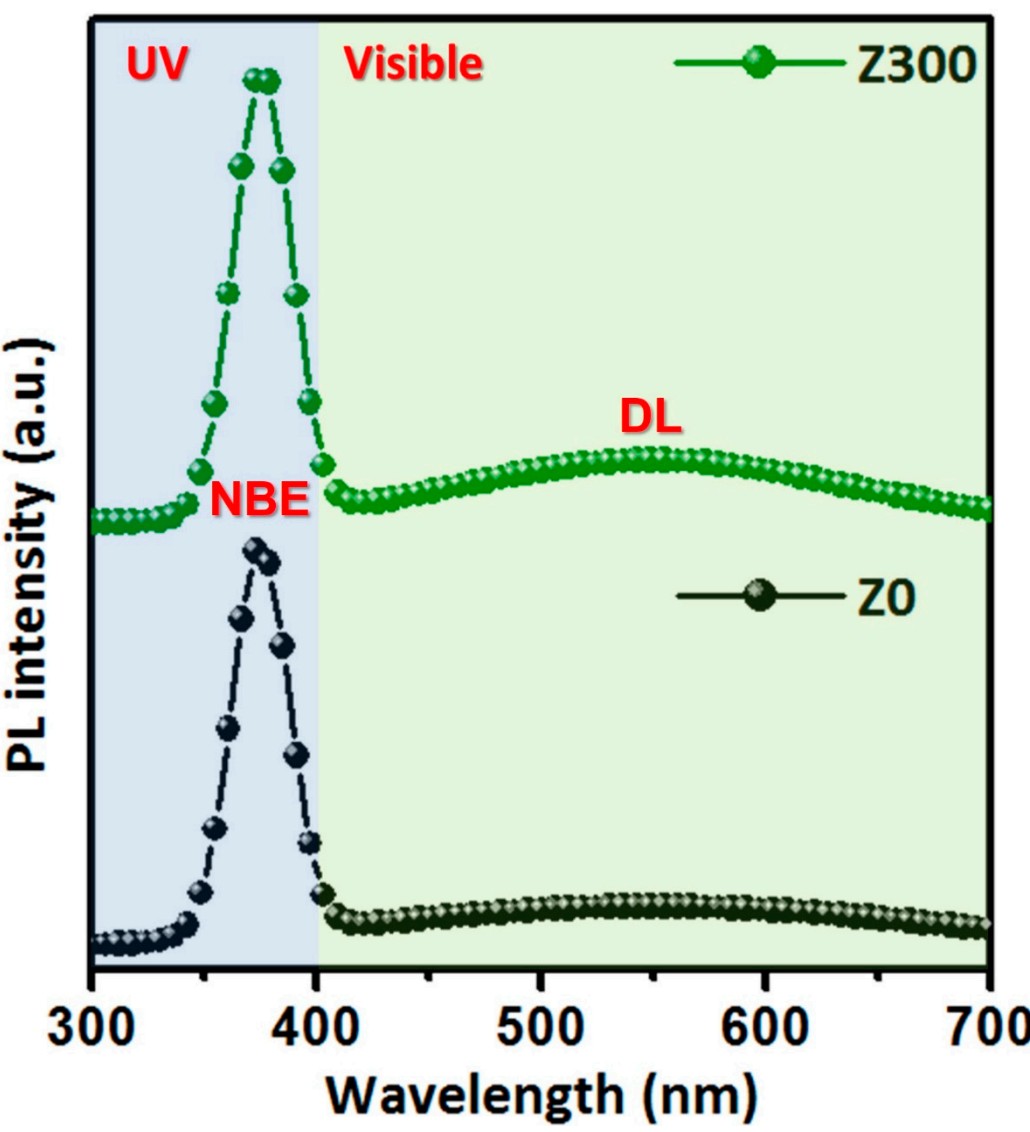

**Figure 5.** Photoluminescence spectra for the samples Z0 and Z300, respectively.

From the above-mentioned results, the highly textured ZnO prepared at an RF power of 75 W without any postannealing modification showed hydrophobic wetting behavior and its corresponding CA value was $95 \pm 1°$. However, photocatalytic behavior has been widely reported and demonstrated by many works [56,57]. With the purpose of modulating the wetting behavior of the ZnO, the sample Z0 was illuminated by ultraviolet (UV) light with a wavelength of 365 nm. This wavelength and strength of the UV light had a photon energy larger than the intrinsic band gap (3.37 eV) of the pristine ZnO. The relationship between the different strengths of the UV illumination ranged from 330 to 3330 mW/cm$^2$ and the water contact angles (CAs) are displayed in Figure 6. Figure 6 displays the related water CA images accompanied by the as-prepared ZnO compound (sample Z0). The drop of liquid water spread on the surface of sample Z0 with a nanograin structure after UV illumination for 10 min with different strengths ranged from 330 to 3330 mW/cm$^2$. It can be clearly observed that the CA value of sample Z0 was reduced from $95 \pm 1°$ to $67 \pm 1°$ under the ultraviolet illumination with 330 mW/cm$^2$ UV power. Moreover, the water CA value was highly reduced from $95 \pm 1°$ to $49 \pm 1°$ under the ultraviolet illumination with 700 mW/cm$^2$ UV strength. Then, the water CA angles were dramatically dropped from $95 \pm 1°$ to $23 \pm 1°$ and $19 \pm 1°$ under the ultraviolet illumination with 1520 and 3330 mW/cm$^2$ UV strength, respectively. This demonstrates that the surface free energy of the ZnO compound was significantly modulated by the different strengths of ultraviolet illumination and demonstrated the typical photocatalytic characteristics. In addition, the rapid reduction in the CA values of the as-prepared ZnO compound was contributed from the photocatalytic action in the ZnO, owing to the generated electron-hole pairs upon photoillumination induced by photoelectron emission. These holes and electrons can either recombine or move to the surface of semiconducting oxides to react with groups adsorbed on the surfaces of the semiconducting oxides. Furthermore, when the UV light was illuminated on the ZnO compound and the generated photon energy was equal to or greater than the band gap of the ZnO compound, the electrons (e$^-$) were excited into the conduction band from the valence band. Simultaneously, the same number of holes (h$^+$) were produced in the valence band and some of the holes reacted with surface oxygen atoms or lattice oxygen (O$^{2-}$) to develop surface oxygen vacancies O$^{1-}$, while the electrons combined with lattice metal ions (Zn$^{2+}$) to develop Zn$^{2+}$ defective sites. With the purpose of absorbing dissociatively on the imperfect sites, the oxygen and water molecules would contest with each other. The surface status of the semiconducting oxides caught electrons (Zn$^+$) toward reacting with oxygen molecules adsorbed on the surface status of the semiconducting oxides. At the same time, the water molecules could coordinate with oxygen vacancy sits (V$_O$), which led to the dissociative adsorption of the water molecules onto the surface status of the ZnO semiconducting oxide. The imperfect sites were kinetically more desirable for the hydrophilic adsorption of hydroxyl groups (OH$^-$) than the adsorption of oxygen molecules.

In addition, Equations (3)–(7) depict the photocatalytic mechanism of the ZnO semiconducting oxide with imperfect sites on its surface.

$$ZnO + 2h\nu \rightarrow 2h^+ + 2e^- \tag{3}$$

Most of the lattice oxygen (O$^{2-}$) or surface oxygen atoms combined with the holes to form surface oxygen vacancies O$^{1-}$, while some of the lattice metal ions (Zn$^{2+}$) combined with electrons to form Zn$^{2+}$ imperfect sites, as illustrated in the subsequent equations.

$$O^{2-} + h^+ \rightarrow O^{1-} \text{ (surface hole trapping)} \tag{4}$$

$$Zn^{2+} + e^- \rightarrow Zn^+ \text{ (surface electron trapping)} \tag{5}$$

$$O^{1-} + h^+ \rightarrow 1/2 O_2 \text{ (gas)} + V_O \text{ (surface oxygen vacancies)} \tag{6}$$



The water molecules and oxygen could contest with each other to absorb dissociatively on the imperfect sites. The surface-caught electrons (Zn+) could combine with the oxygen molecules adsorbed onto the ZnO surface of the semiconducting oxide:

$$Zn + O_2 \rightarrow Zn^{2+} + O^{2-} \tag{7}$$

UV illumination can modulate the ZnO semiconducting oxide and its related chemical surface states, and therefore manipulate its surface wetting behavior. The above-mentioned results indicate that the surface wetting behavior can be simply modulated by a simple factor, such as the strength of the UV illumination. Besides those mentioned, ZnO is considered as a candidate with many potential devices such as biosensors, biomedical devices, and other multifunctional detecting devices [58–62].

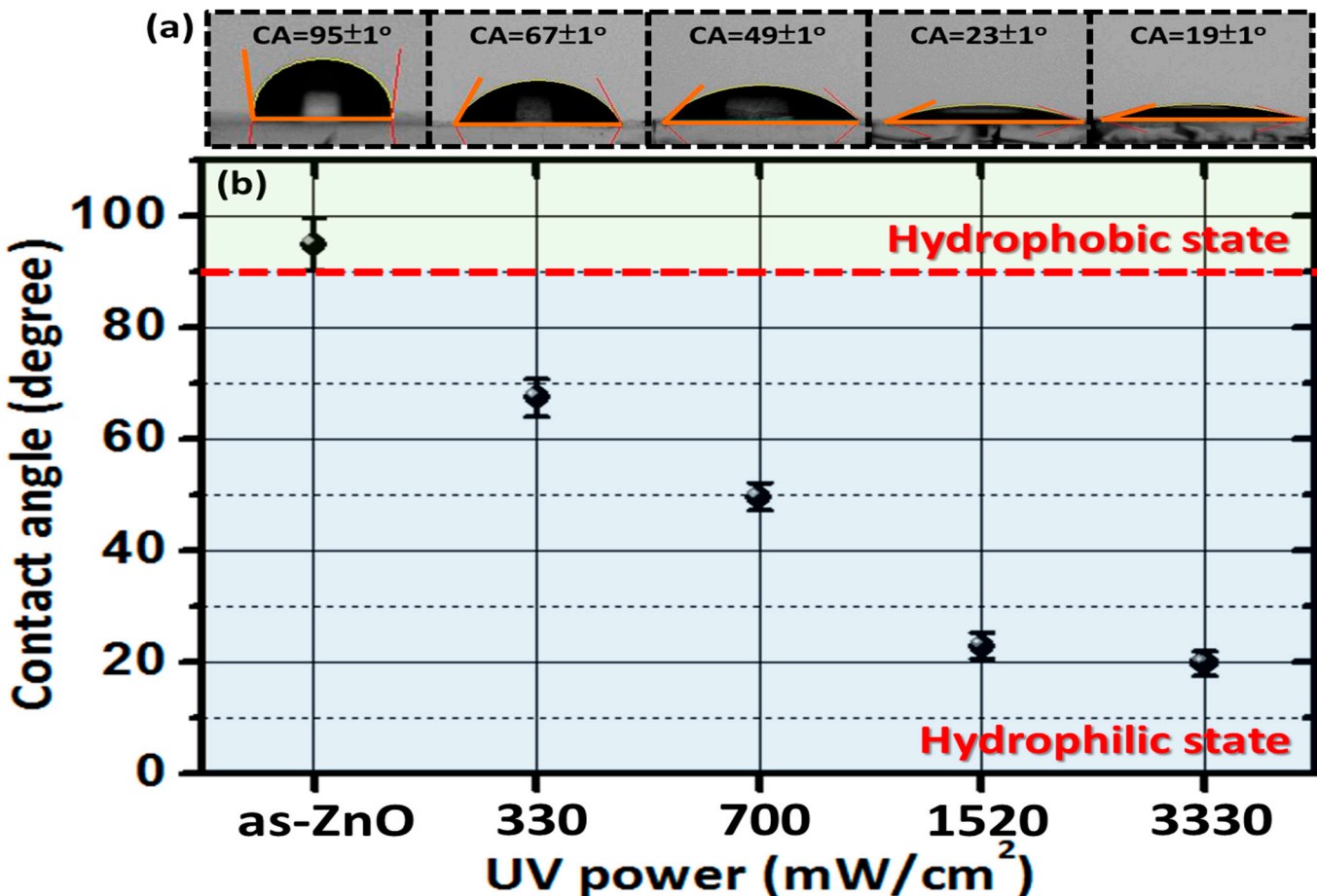

**Figure 6.** (**a**) The contact angle (CA) images of the sample Z0 under UV illumination for 10 min without and with the different UV strengths ranging from 330 to 3330 mW/cm$^2$, respectively. (**b**) The UV strength as a function of CA value for sample Z0 under UV illumination with a light wavelength of 365 nm.

The ZnO films were dispersed in a methylene blue (MB) solution in order to measure the photocatalytic efficiency during UV illumination in 10 min steps, as shown in Figure 7. The photodegradation rate (*K*) of the ZnO films can be calculated using the following Equation (8):

$$-\ln\left(\frac{C}{C_0}\right) = Kt \tag{8}$$

where $C_0$ is the UV-Vis absorbance of the solution before illumination (initial MB concentration), C is the UV-Vis absorbance of the solution after illumination (concentration of

MB at time *t*), *K* is the apparent rate coefficient (kinetic degradation rate), and *t* is the total reaction time of the catalysts.

The photodegradation ratios of the MB solution with and without ZnO films ($C/C_0$) versus exposure/reaction time (*t*) under UV light are displayed in Figure 7a. The investigation in the absence of a photocatalyst indicated an insignificant photocatalytic efficiency of the MB, indicating an insignificant self-photolysis of the MB molecules under the UV light illumination. $C_0$ is the initial concentration of MB and C is the MB concentration varied during the exposure time under UV light illumination. The value of $C/C_0$ for the solution with ZnO films decreased with increasing the exposure time under UV light illumination, so $C/C_0$ was directly estimated from the ratio between the above photoabsorbance values. Figure 7a shows that pure ZnO film is useful as a photocatalyst with enough photocatalytic efficiency. The linear relationship of $-\ln(C/C_0)$ as a function of the reaction time is shown in Figure 7b, and the apparent rate coefficient (*K*) for the MB photodegradation was estimated from the pseudo-first-order approximation, which was proportional to the corresponding values of the photoabsorbance. The photodegradation rate (*K*) was estimated as the slope of the linear fit in $\ln(C/C_0)$ versus the *t* plot for all samples, and the kinetic degradation rate (*K*) for the pure ZnO film was equal to $2.93 \times 10^{-3}$ (min$^{-1}$). Therefore, the general guideline of the dye-sensitized photocatalytic reaction corresponding to the pure ZnO film in the MB solution is displayed in Figure 7.

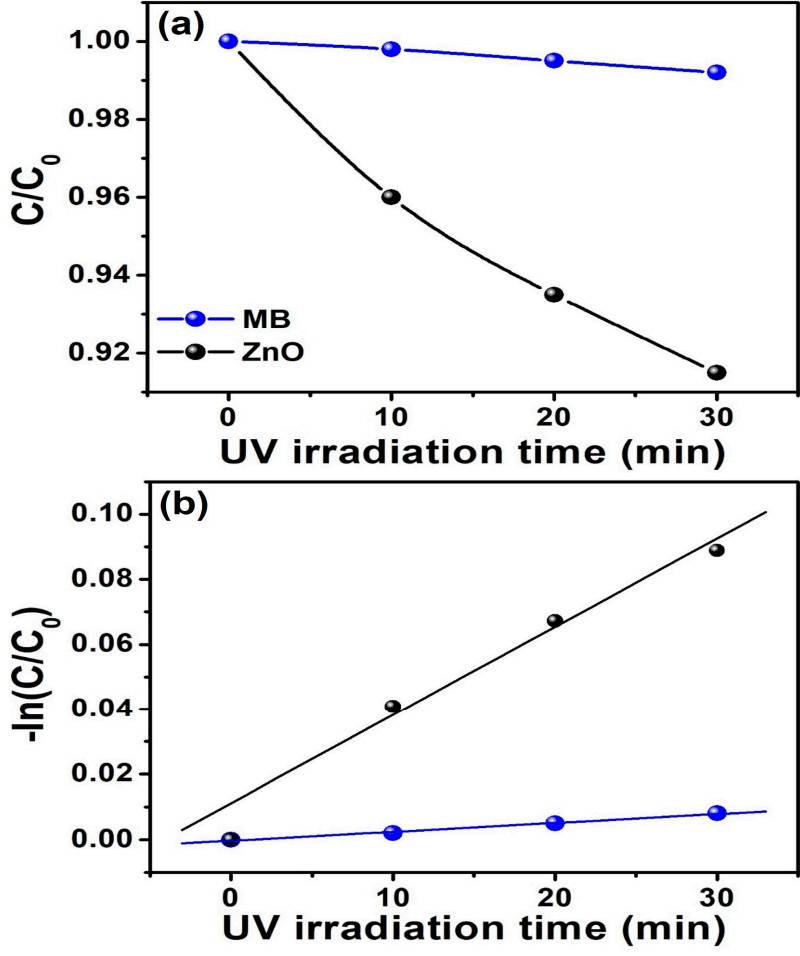

**Figure 7.** (**a**) ($C/C_0$) and (**b**) $-\ln(C/C_0)$ ratios as a function of reaction time for photocatalytic degradation of MB solution in the presence of the ZnO film under UV illumination.

When the absorption of energy is higher than the bandgap energy of the ZnO, the holes ($h_{VB}^+$) are generated in the valence band (VB) and electrons ($e_{CB}^-$) in the conduction band (CB), respectively. This phenomenon was directly related to the existence of super-

ficial ZnO lattice oxygen atoms/ions that could be associated with the oxidation of the adsorbed MB molecules, leading to the existence of surface hydroxyl groups acting as hole-acceptors to generate OH· radicals, which can be responsible for the generation of superoxide ions [63–68]. Thus, the hydroxyl radicals (·OH) decomposed the MB effectively to generate water ($H_2O$), carbon dioxide ($CO_2$), and other inorganic by-products.

To further examine the plasmonic coupling efficacy, a 15 nm thick Pt nanolayer was selected to be deposited onto the ZnO compound using direct-current (dc) sputtering, which was denoted as Pt/ZnO. A Pt thin film with a continuous layered structure was deposited onto a ZnO semiconducting oxide, as observed in the two-dimensional (2D) AFM topographical morphology displayed in Figure 8a. The continuous growth of the capping nanolayer was also checked using a three-dimensional (3D) AFM micrograph, as displayed in Figure 8b. From the AFM line-scan displayed in Figure 8c, the height of the Pt continuous structure was observed. The plan-view SEM image of the Pt/ZnO heterostructures showed a highly uniform state, and typically displayed a flat surface status, as displayed in Figure 8d. It can be seen that the CA value of the Pt/ZnO heterostructures was changed from $95 \pm 1°$ (ZnO) to $86 \pm 1°$ (Pt/ZnO) while capping with a Pt continuous nanolayer, as displayed in Figure 8e. X-ray scattering was used to ideally complement these microscopic methods, since it could successfully reflect the representative microstructural information for a large sample area of the Pt/ZnO heterostructured films, as displayed in Figure 8f. The XRD pattern for the Pt/ZnO heterostructured films showed typical ZnO (002) and Pt (111) diffraction peaks, which supports the presence of a Pt nanolayer coated onto a ZnO semiconducting oxide, which can be indexed to Pt/ZnO bilayer films in a crystalline phase. In addition, no diffraction peak of an indefinite phase from the Pt/ZnO bilayer films existed, thus suppling strong proof for the coverage of the ZnO by the Pt without an intermixed phase. The above structural and morphological measurements support the configuration of the Pt/ZnO heterostructured films, and the XRD spectra measurement was consistent with the observation from the topical AFM and SEM images, as displayed in Figure 8.

In addition, the surface plasmonic resonance (SPR) was induced by coupling oscillations between the electrons and photons that could be obtained and generated from the interface between the Pt metal and ZnO semiconductor, with the purpose of designing SPR heterostructures. Usually, the pure metals could offer surface plasmonic collective oscillations of free electrons, which could collect the electromagnetic (EM) waves to a little fraction of wavelength while enhancing the local field energy by several orders of magnitude [69]. It was reported that the local EM field known by some molecules on the metal surface at the nanoscale was dramatically enhanced, producing a considerably enhanced Raman intensity [70]. We concluded that such a kind of ZnO semiconductor capped with a continuous Pt nanolayer with a suitable thickness could induce a localized field onto its surface, producing a coupling effect and displaying the SPR. With the purpose of confirming this idea, the Raman spectra of the pristine ZnO and Pt/ZnO bilayer heterostructures at room temperature were measured, as shown in Figure 9. The Pt/ZnO heterostructured films showed an outstanding signal enhancement, and the frequency of the 1LO phonon peak was positioned at about 573 $cm^{-1}$ corresponding to the $A_1$ (LO) phonon, while the $E_1$ (TO) mode was forbidden owing to the Raman selected rule [71,72]. Thus, the absence in the measurements with the TO mode further confirmed that the ZnO film had a high *c*-axis orientation. Hence, all the measured results inferred from the X-ray diffraction patterns and Raman analysis showed the same tendency.

The above result also demonstrated surface-enhanced Raman scattering that could be gained by capping with a Pt continuous nanolayer onto a ZnO semiconducting oxide. Therefore, a simple method is presented and claimed here that the surface plasmonic resonance and surface wetting behavior of ZnO compounds can be effectively modulated by capping a Pt continuous nanolayer, compared to be decorated with metallic nanoparticles. The Pt/ZnO heterostructures exhibited some novel and valuable employments in the near

future due to their multifunctional material characteristics acting as a typical structure of nanodevice design.

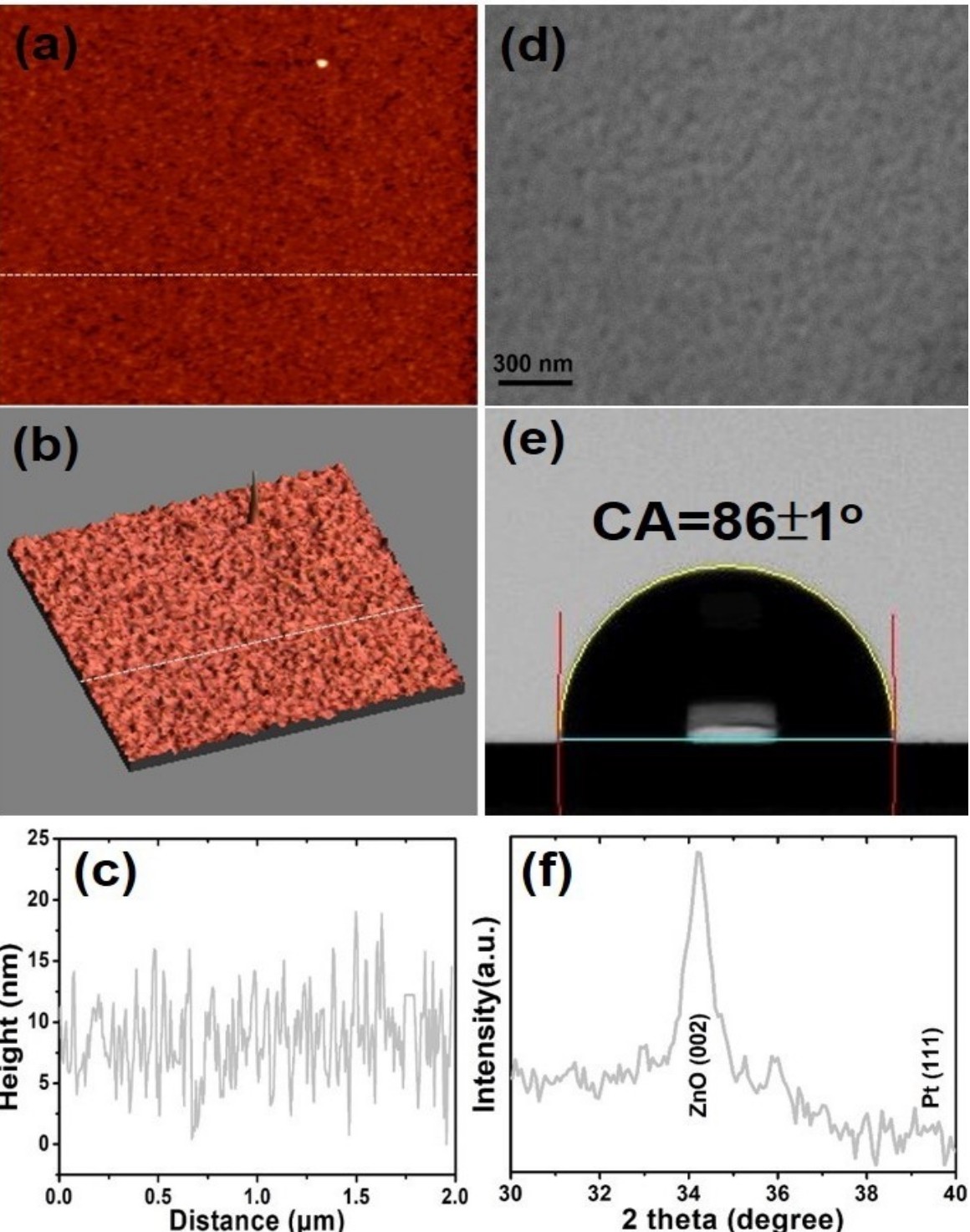

**Figure 8.** AFM topographical morphology for the Pt/ZnO bilayer films (**a**) 2D feature; (**b**) 3D feature for (**a**), and (**c**) a line scan profile of Pt/ZnO bilayer films related to (**a**,**b**), with a scanning area of $2 \times 2\ \mu m^2$. (**d**) A topical SEM micrograph. (**e**) Related CA image. (**f**) Related X-ray diffraction pattern.

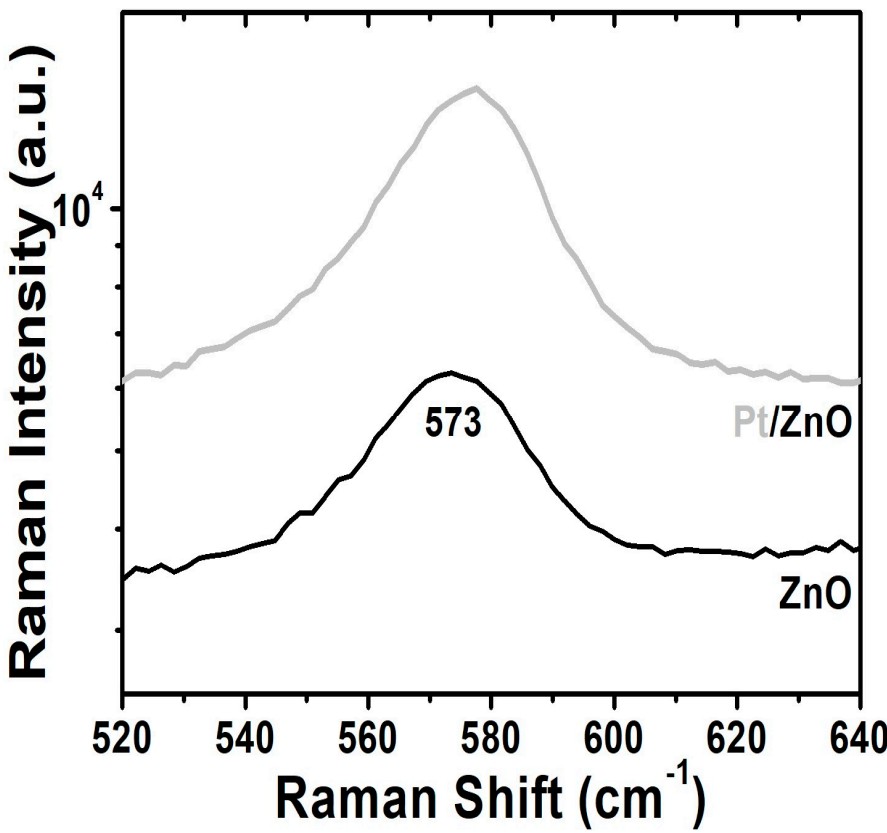

**Figure 9.** Raman spectra for pristine ZnO semiconducting oxide without and with a Pt capping nanolayer measured at room temperature.

## 4. Conclusions

In this present work, textured ZnO films were directly prepared onto glass substrates with a (002) orientation by applying a radio-frequency magnetron sputtering system without supplying an oxygen source at room temperature, and an even epitaxial ZnO (004) facet could be obtained via a postannealing modification at 300 °C. All the ZnO semiconducting oxides demonstrated an excellent crystallinity and high level of visible transmittance (beyond 80%). The as-prepared ZnO semiconducting oxide displayed a hydrophobic status, which was then transferred to a hydrophilic status via the simple postannealing modification. The CA value, internal stress, and optical transmittance of the photoluminescence for the ZnO semiconducting oxide were significantly modulated by the surface microstructure, which was induced by a simple factor of the postannealing modification. The rapid CA conversion for the ZnO semiconducting oxide could be modulated from hydrophobic (95°) to superhydrophilic (19°) by using a simple method via different strengths of UV light illumination. The rapid transformation of the CA for the ZnO semiconducting oxide was owed to the photocatalytic reaction caused by the positive surface charges accumulated on the surface via photoelectron emission. A greater impact on the photocatalytic efficiency of the ZnO film for the MB degradation was confirmed. Therefore, a simple way was proposed herein that the CA and photocatalytic variation of ZnO compounds could be directly modulated by a postannealing modification and ultraviolet illumination, respectively. The enhanced Raman intensity in the Pt/ZnO bilayer heterostructures could be obtained, owing to the strong coupling of the light emission with the localized surface plasmonic resonance of the Pt nanolayer, which could generate an effectual way of leading to more photonic scattering from incident light. This could further be used for sensing and the multifunctional nanodevice design of wearable technologies.

**Author Contributions:** Conceptualization, D.-H.W.; methodology, D.-H.W.; software, D.-H.W., S.-K.T. and S.-C.C.; validation, D.-H.W.; formal analysis, D.-H.W.; investigation, D.-H.W.; resources, D.-H.W. and R.-T.H.; data curation, D.-H.W. and S.-C.C.; writing—original draft preparation, D.-H.W.; writing—review and editing, D.-H.W.; visualization, D.-H.W.; supervision, D.-H.W.; project administration, D.-H.W. and R.-T.H.; funding acquisition, D.-H.W. and R.-T.H. All authors have read and agreed to the published version of the manuscript.

**Funding:** This work was funded by the National Science and Technology Council (NSTC) through Grants numbered 112-2740-M-027-001 and 112-2221-E-027-111 and the University System of Taipei Joint Research Program through Grant No. USTP-NTUT-NTOU-111-02. The authors acknowledge resources and support from the Quantum Materials Shared Facilities of Institute of Physics at Academia Sinica.

**Institutional Review Board Statement:** Not applicable.

**Informed Consent Statement:** Not applicable.

**Data Availability Statement:** Data are contained within the article.

**Conflicts of Interest:** The authors declare no conflict of interest.

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
