# Peer review of "The Surface Behavior of ZnO Films Prepared at Room Temperature"

_jcs, doi:10.3390/jcs7080335_

Round 1

Reviewer 1 Report

Please refer to the enclosed comment file.

Moderate changes are needed. 

Author Response

I have concluded reviewing the manuscript. In this study, ZnO films have been prepared via room temperature radio-frequency magnetron sputtering by varying different deposition parameters, and the microstructures, optical properties, photocatalytic, etc. properties were characterized. Although the study is well executed, I noticed the presentation of the results needs improvement. Please refer to the comment file for more information. Please address the following issues:

  • The following sentence is unclear: “The functional coatings deposited on metal, ceramics, semiconductor, and corresponded nanostructures are easy and useful way to obtain significantly surface plasmonic enhancement in the electrical and optical performances combined with bottom materials” What is bottom material. Rephrase the above sentence.

Our response:

Thanks for the Reviewer’s comments. The rephrase sentence has been modified in red characters in the revised manuscript and listed below.

Functional coatings can be deposited on metal, ceramics, and semiconductor bottom-based materials, and their bottom-up nanostructures are an easy and useful way to significantly enhance surface plasmonic performance in the electrical and optical applications.

(2) Authors wrote: “………iolet (UV) light irradiation owning to its wide band gap”. Include the value of band gap and references.

Our response:

Thanks for the Reviewer’s kind reminder. The value of ZnO band gap (3.37 eV) and related reference [21] have been modified in red characters in the revised manuscript.

(3) Authors wrote: “The glass substrates in chamber have been placed parallel to ZnO binary target with 99.99% purity, whose thickness and diameter are 3 mm and 51 mm, respectively.” Describe how the target was prepared. Was it obtained commercially?

Our response:

Thanks for the Reviewer’s kind reminder. The ZnO binary target with 99.99% purity is obtained from commercial sources. The text “commercial” has been added in the revised manuscript.

(4) Authors wrote: “……were cleaned in acetone and ethanol at first to withdraw” Use remove instead of withdraw.

Our response:

Thanks for the Reviewer’s comments. The text “remove” has been used instead of withdraw in the revised manuscript.

(5) Rewrite the following. It is not clear. “Furthermore, for the purpose of obtaining modulable grain microstructures, one of the as-prepared ZnO sample was during postannealing modification in furnace at 300°C for 30 minutes after deposition.”

Our response:

Thanks for the Reviewer’s comments. The sentence has been modified in red characters in the revised manuscript and listed below.

Furthermore, for the purpose of obtaining modulable grain microstructures, the as-prepared ZnO samples (Z0) were treated with a postannealing modification in furnace at 300°C for 30 minutes after deposition (Z300).

(6) Provide reference to the equation: σ = 450(c-c0)/c0 GPa

Our response:

Thanks for the Reviewer’s kind reminder. The related reference [50] have been added in red characters in the revised manuscript.

(7) Replace the following It can also be examined that the apertures located between grain and grain can catch oxygen inside and exist onto the surface area of top film. With It can also be due to the presence of pores between the two grains that can absorb oxygen that exists on the surface of the film (or something like that).

Our response:

Thanks for the reviewer’s suggestion. The sentence has been modified in red characters in the revised manuscript and listed below.

It can also be due to the presence of apertures or pores located between the grains that can absorb oxygen inside and exist onto the surface of film.

(8) Authors wrote: “The modulable wetting behavior was obtained by adjusting the strength of UV light irradiation and storage in dark room for 12 hrs to reach the initial water contact angle.” 2 Please describe what changes we expect by storing the film in a dark room.

Our response:

Thanks for the Reviewer’s kind reminder. The modulable wetting behavior was owning to photocatalytic reaction, which was contributed from the generating electron-hole pairs by UV light irradiation, and the water molecules and oxygen would contest with each other in dark. The described sentences have also been modified in red characters in the revised manuscript.

(9) Authors wrote: “It can be understood that the deposited uniformity of nanostructures, thin, and thick films with super large area was caused by various sputtering specifications including sputtering power, sputtering pressure, supplying content gas, substrate temperature, and the distance between target and substrate.” The following two article discuss extensively on the modulation of microstructure using various deposition parameters. Inclusion of these two reference will help the readers and also provide some good background on film preparation. They also cover ZnO films. https://doi.org/10.1016/j.jmat.2020.02.008

https://doi.org/10.1016/j.mtener.2022.100965

Our response:

Thanks for the reviewer’s suggestion. The recommended good references [48, 49] have been added in the revised manuscript. [48] Chen, X.; Zhou, Z.; Lin, Y.H.; Nan, C. Thermoelectric thin films: Promising strategies and related mechanism on boosting energy conversion performance. J. Materiomics, 2020, 6, 494-512. [49] Nandihalli, N. Thermoelectric films and periodic structures and spin Seebeck effect systems: facets of performance optimization. Mater. Today Energy, 2022, 25, 100965.

(10) Finally, I noticed several problems in the language. Please make the language more academic.

Our response:

Thanks for the reviewer’s suggestion. All described sentences have also been modified in red characters in the revised manuscript.

Reviewer 2 Report

The paper reads well and is interesting.  Some comments:

I expected that differential sputtering would make a Zn rich film. Do the authors know final stoichiometry? I assume that the oxygen content changes with heating. 

How thick are these films?  Have the authors looked at delamination or potentially a simple tape test?

I think there is something funny with equation 7.  I do not think the authors mean to imply that there is metallic zinc on the surface.

I think the Pt layer is too thin for an XRD.  I do not see a peak (the Pt(111) diffraction peak).  It is consistent with a Pt overlayer as mentioned in the paper. I believe that they made a Pt overlayer. I just do not think the XRD is the proof.

Maybe label the maximum for both peaks in Figure 9.

Figure 2.  Is this figure worth it?  Table or just in the test.

Author Response

The paper reads well and is interesting. Some comments:

I expected that differential sputtering would make a Zn rich film. Do the authors know final stoichiometry? I assume that the oxygen content changes with heating.

Our response:

Thanks for the reviewer’s comment. The chemical stoichiometry of the binary ZnO thin films with a thickness of about 200 nm was identified to be Zn40O60 by field emission electron probe X-ray microanalysis as shown in the Ref. fig.1. The described sentence has been modified in red characters in the revised manuscript.

Ref. fig. 1 The chemical stoichiometry of the binary ZnO thin films has been investigated by EPMA for the thickness ranged from 100 to 300 nm.

  • How thick are these films?  Have the authors looked at delamination or potentially a simple tape test?

Our response:

The single ZnO films have been prepared using 75 W RF power, and all the ZnO film thickness was fixed at about 250 nm as shown in the Ref. fig.2. Due to the Pt layer is too thin (15 nm thick), the height of Pt continuous structure was checked from AFM line-scan as displayed in Figure 8c, The plan-view SEM image of Pt/ZnO heterostructures showed uniform state, and typically displayed flat surface morphology as displayed in Figure 8d.

Ref. fig. 2 The cross-sectional SEM image for pure ZnO film.

  • I think there is something funny with equation 7.  I do not think the authors mean to imply that there is metallic zinc on the surface.

Our response:

Thanks for the reviewer’s comment. We agree with reviewer’s point of view. We just want to describe that the surface of any type ZnO will catch electrons (Zn+) toward to combine with oxygen molecules adsorbed on the ZnO surface.

  • I think the Pt layer is too thin for an XRD. I do not see a peak (the Pt(111) diffraction peak).  It is consistent with a Pt overlayer as mentioned in the paper. I believe that they made a Pt overlayer. I just do not think the XRD is the proof.

Our response:

Thanks for the reviewer’s comment. We agree with reviewer’s point of view. The X-ray photoelectron spectroscopy (XPS) analysis will be checked and discussed in the next surface plasmon resonance experiments.

  • Maybe label the maximum for both peaks in Figure 9.

Our response:

We fully thank for the Reviewer’s kind reminder and suggestion. The label maximum for the Raman spectra measurement has been shown in revised Figure 9.

(revised) Figure 9. Raman spectra for pristine ZnO semiconducting oxide without and with Pt capping nanolayer measured at room temperature.

  • Figure 2.  Is this figure worth it?  Table or just in the test.

Our response:

Thanks for the Reviewer’s kind reminder. The related values for ZnO samples (Z0) and (Z300) have been marked in red characters in the revised manuscript. This will help readers easily understand.

Round 2

Reviewer 1 Report

The amended changes are satisfactory and therefore I recommend accepting the manuscript for publication.

Reviewer 2 Report

The paper is acceptable for publication. Thanks for considering my comments.